# Macrophage Polarization in Heterotopic Ossification: Inflammation, Osteogenesis, and Emerging Therapeutic Targets

**DOI:** 10.3390/ijms26125821

**Published:** 2025-06-17

**Authors:** Yifei Ren, Wenwen Zhao, Mengchao Liu, Hui Lin

**Affiliations:** 1School of Basic Medical Sciences, Jiangxi Medical College, Nanchang University, Nanchang 330006, China; jp4217121030@qmul.ac.uk (Y.R.); jp4217123200@qmul.ac.uk (W.Z.); 18336190623@163.com (M.L.); 2Queen Mary School, Jiangxi Medical College, Nanchang University, Nanchang 330006, China

**Keywords:** heterotopic ossification, macrophages, polarization, mechanism, therapy

## Abstract

Heterotopic ossification (HO) refers to an abnormal process characterized by the aberrant development of bone within soft tissues, leading to significant impairments in patients’ mobility and overall quality of life. Macrophages, as a crucial element of the immune system, are instrumental in the different stages of heterotopic ossification through their dynamic polarization state (pro-inflammatory M1 and anti-inflammatory M2 phenotypes) and secretion of different cytokines. This review explores novel mechanisms of M1 and M2 macrophage-mediated heterotopic ossification, emphasizing the involvement of the inflammatory microenvironment, osteogenic factors, and osteogenic signaling pathways. In addition, we explore promising therapeutic strategies targeting macrophage polarization and function, including agents that modulate the inflammatory microenvironment, such as IL-1 inhibitors, parovastatin, and metformin, as well as agents that affect macrophage osteogenic signaling, such as TGF-βRII-Fc, Galunisertib, and Ruxolitinib. A more comprehensive understanding of these mechanisms may open up new avenues for developing novel approaches to reducing HO in high-risk patients.

## 1. Introduction

Heterotopic ossification is a pathological process that results in the formation of ectopic bone outside of the skeleton, including soft tissues and joints [1], and can be divided into two main categories: hereditary HO and acquired HO (Figure 1) [2]. Fibrodysplasia ossificans progressiva (FOP) is the most common hereditary HO and belongs to a group of rare hereditary disorders, with an incidence of approximately 1 case per 2 million people [2]. The cause of FOP is closely related to mutations in the ACVR1 gene, of which the most common mutation is the R206H mutation [3,4]. This gene encodes the bone morphogenetic protein (BMP) type I receptor, and its mutation results in abnormal activation of the BMP signaling pathway, which triggers the pathological changes in FOP [5]. FOP mainly manifests as the HO of muscles, ligaments, and other connective tissues [6], and patients usually present in childhood with inflammatory soft tissue swelling followed by progressive formation of heterotopic bone [5]. Although patients with FOP are usually normal at birth, minor trauma or inflammation (e.g., vaccinations or viral infections) is often seen as a trigger for an episode of HO [2,5]. Another autosomal dominantly inherited form of heterotopic ossification is progressive ossification heterotopia (POH), which is characterized by the formation of intramembranous ectopic bone tissue and is not significantly associated with an inflammatory response [7]. Acquired HO is usually triggered by musculoskeletal trauma or central nervous system (CNS) injury [2]. Traumatic heterotopic ossification (THO) occurs after soft tissue injury (severe burns, fractures, etc.) [1], and is a process of heterotopic bone formation within the cartilage caused by a complex inflammatory response and dysregulation of osteogenic factors, although its pathogenesis is unclear [2]. Studies have shown that the prevalence of neurogenic heterotopic ossification (NHO), a complication of CNS injury (spinal cord injury (SCI), traumatic brain injury (TBI), etc.), is approximately 20% [8]. The pathological process of NHO is still not fully elucidated, and available studies suggest that its occurrence involves multiple mechanisms such as osteogenic factors, neuroinflammation, and macrophages [2].

During the development of heterotopic ossification, inflammatory response, chondrogenic differentiation, osteogenic differentiation, and ectopic osteogenesis are the four key steps [9]. After the early inflammatory response, both acquired HO and FOP utilize endochondral ossification as the mechanism for ectopic bone formation [1]. Unlike POH, endochondral ossification involves an intermediate cartilage stage [1]. In this phase, mesenchymal stem cells differentiate into chondrocytes and begin producing the cartilage matrix [10]. A perichondrium forms around these newly generated chondrocytes, establishing the boundary for skeletal development [10]. These cells exit the cell cycle, undergo hypertrophy, and trigger the differentiation of osteoblast progenitors within the perichondrium [11]. Chondrocyte hypertrophy is a critical step in osteoblast differentiation. As blood vessels invade the hypertrophic cartilage, osteoblast differentiation is further activated [10]. Eventually, hypertrophic chondrocytes undergo apoptosis, while the remaining chondrocytes differentiate into osteoblasts, driving bone formation [10].

## 2. Macrophage Polarization in HO

As central cells in the immune system, macrophages not only participate in inflammatory responses and tissue remodeling, but also play roles at different stages of HO through various mechanisms [12]. During the initial stages of HO, macrophages become activated and recruited to the injury site, where they play a role in creating an inflammatory and hypoxic microenvironment while regulating the chondrogenic and osteogenic differentiation capacities of MSCs [9,13]. Macrophages in HO include two types: resident macrophages, which respond rapidly to injury, and macrophages differentiated from circulating monocytes, which quickly differentiate into inflammatory macrophages upon receiving stress signals, becoming the primary initiators of the inflammatory response [14,15]. According to the activation state, macrophages can be categorized into two subtypes, M1 and M2 [16].

Due to the remarkable heterogeneity and plasticity of M2 macrophages, various stimuli can induce their polarization into functionally distinct subtypes [17]. Currently, four major M2 subtypes have been identified: M2a, M2b, M2c, and M2d (Figure 1) [17]. M2a macrophages are polarized in response to IL-4 or IL-13 stimulation and are characterized by the expression of CD206 and arginase-1 [18,19]. Through the secretion of factors such as IL-10 and TGF-β, M2a macrophages contribute to inflammation resolution, tissue repair and remodeling, and angiogenesis [18,20]. M2b macrophages are polarized upon exposure to immune complexes, TLR signaling, and IL-1 receptor activation. They are notable for their simultaneous production of pro-inflammatory cytokines and high levels of IL-10 [20]. M2c macrophages are induced by IL-10, TGF-β, and glucocorticoids, as well as express markers such as CD163, MMP9, and CD206 [20,21]. These macrophages secrete abundant IL-10 and TGF-β and are involved in the clearance of apoptotic cells [22]. M2d macrophages, often referred to as tumor-associated macrophages, are primarily induced by IL-6 and participate in immunosuppression and angiogenesis [18,23]. Functionally distinct M2 subtypes play critical roles in tissue remodeling following bone injury. In fracture hematoma and the adjacent bone marrow, M2a, M2b, and M2c macrophages have been detected, although their precise mechanisms of action remain incompletely understood [23]. We hypothesize that heterotopic ossification, as an injury-induced process of endochondral bone formation, may similarly involve M2 macrophages exhibiting such functional heterogeneity.

Research has shown that M1 and M2 macrophages infiltrate and function differently at various stages of HO. M1 macrophages infiltrate and become active early after injury (e.g., 1 day post-injury (dpi)), primarily secreting pro-inflammatory cytokines that stimulate abnormal inflammatory responses in the HO immune microenvironment and promote the proliferation of HO-initiating cells. In contrast, M2 macrophages significantly increase in number by 3 dpi, exhibiting immunosuppressive properties and secreting anti-inflammatory factors and osteogenic factors (e.g., TGF-β, VEGF) to facilitate tissue remodeling and angiogenesis [12,24]. The microenvironment created by M2 macrophages supports chondrogenesis and osteogenic differentiation of HO-initiating cells (e.g., MSCs) while promoting tissue remodeling [12]. A recent spatial transcriptomics study further demonstrated the dynamic alterations of macrophages within the inflammatory microenvironment of HO [25]. At 1 dpi, M1 macrophages engage in extensive interaction with MSCs, facilitating their transition from a quiescent to a proliferative state; by 3 dpi, M1-derived M2 macrophages promote the chondrogenic differentiation capacity of MSCs [25]. Based on these findings, suppressing the early inflammatory response (1 dpi) and reversing late-stage immunosuppression (3 dpi) may represent promising intervention strategies for the treatment of HO.

Although the specific mechanisms of macrophage polarization and its subtypes in HO have not been fully elucidated, their complex roles in HO development are widely recognized. The polarization states of macrophages and their secreted cytokines directly regulate bone healing and tissue repair processes, offering new directions for HO treatment.

## 3. Mechanism of Macrophage in HO

### 3.1. Regulation of Inflammatory Microenvironment

It is commonly recognized that bone remodeling and repair are significantly influenced by the immune system. The inflammatory microenvironment influences the formation and progression of HO in traumatic HO [26] and FOP [27], while the concentrations of pro-inflammatory cytokines are positively correlated with the incidence of HO [28]. As key mediators of immune responses and major sources of cytokines in bone regeneration [29], macrophages play a crucial role in establishing the inflammatory microenvironment [26]. Following inflammatory and traumatic stimulation in the early stages, various immune cells, including macrophages and monocytes, infiltrate the injury site, leading to the accumulation of inflammatory cytokines such as tumor necrosis factor (TNF), interleukin-1β (IL-1β), IL-6, and monocyte chemoattractant protein-1 (MCP-1) at the injured site. This induces local inflammatory responses and tissue repair while enhancing the osteogenic potential of mesenchymal cells [14,26,30,31]. The infiltration of macrophages and increased inflammatory cytokines disrupt tendon homeostasis, favoring permissive niches for HO formation, promoting stem cell recruitment, and activating osteogenic signaling pathways, ultimately leading to chondrogenesis and the progression of HO [30,31,32].

The complex regulatory mechanisms of macrophages with different polarization states in the inflammatory microenvironment of HO are still under investigation. During the initial inflammatory response of THO, M1 macrophages infiltrate and secrete high levels of pro-inflammatory cytokines (e.g., IL-1β, IL-6, and TNFα), whose expression increases with M1 macrophage infiltration [9].

Matsuo et al. compared FOP-M1-type induced pluripotent stem cell-derived macrophages (iMACs) with wild-type M1-type iMACs and found that FOP-M1-type iMACs secreted higher levels of IL-6 and TNF-α without stimulation, significantly improving the inflammatory microenvironment of HO. This suggests that M1 macrophages exhibit an aberrant pro-inflammatory response, potentially providing a background for continuous immunostimulation in FOP patients [33].

In contrast, M2 macrophages exhibit immunosuppressive and anti-inflammatory functions, with several subtypes such as M2a and M2b playing key roles in immune regulation [22,24]. During bone regeneration, M2a macrophages activated by IL-4 and IL-13 secrete various anti-inflammatory factors, creating a favorable immune microenvironment. These M2 macrophages promote osteogenic differentiation by regulating the BMP-2/SMAD signaling pathway in MSCs [34]. In addition, the immune checkpoints (ICs) expressed by macrophages play a central role in regulating immune homeostasis in HO [35].

In summary, the immune system plays a central role in HO formation by regulating the microenvironment. As pivotal elements in the immune system, macrophages modulate immune responses and influence the development of HO through the secretion of cytokines like IL-6, IL-1, TNF-α, and IL-10. The involvement of immune checkpoint proteins and immune regulatory factors further enriches the complexity of this pathological process (Figure 2).

#### 3.1.1. Pro-Inflammatory Cytokines

IL-6

IL-6 is a key pro-inflammatory cytokine secreted by M1 macrophages. During the initial phases of HO, its expression level increases in parallel with M1 macrophage infiltration, contributing to the formation of the inflammatory response [9].

Oncostatin M (OSM), belonging to the IL-6 family, is produced by a variety of immune cells (e.g., neutrophils [36], macrophages [37], monocytes [38]), as well as BM stromal cells and osteoblasts during the initial inflammatory phase, and is involved in various stages of induced cellular osteogenic differentiation [39]. The OSM receptor (OSMR) is widely present in osteoblasts, osteocytes, and other cell types. In the osteogenic microenvironment, OSM interacts with the transmembrane glycoprotein 130 (GP130), which subsequently recruits the OSMR, thereby triggering the activation of the JAK/STAT signaling pathway [39]. Both in vivo and in vitro studies have demonstrated that during the inflammatory phase of bone healing following injury, OSM released by M1 macrophages activate the OSM/OSMR/JAK/STAT3 signaling pathway, thereby promoting the osteoblast differentiation and maturation of BMSCs [40]. Furthermore, studies have revealed that OSM expression is significantly upregulated in the heterotopic ossification model [41]. In the SCI-induced NHO mouse model, OSM is secreted by infiltrating macrophages and activates the JAK1/2-STAT3 signaling pathway via the GP130-OSMR complex, leading to STAT3 phosphorylation and accelerating the progression of NHO [41].

IL-1β

IL-1β, as an M1 macrophage-associated cytokine, is an important marker for determining M1 polarization and plays a crucial role in the initiation and progression of HO [42]. Immunohistochemical analysis of human NHO biopsies revealed an abundance of CD68+ macrophages at HO lesion sites, with their distribution closely overlapping that of IL-1β-secreting cells [43]. In addition, the volume of NHO was significantly reduced in IL1r1-/- mice compared with wild-type mice, suggesting that the IL-1 signaling pathway plays a key role in the development of SCI-NHO [43]. IL-1 family members, IL-1α and IL-1β, interact with the same receptor (IL-1R1) and synergistically promote the mineralization process of osteoblasts and fibro-adipose progenitors (FAPs) in vitro [43,44]. However, the lack of either IL-1α or IL-1β alone does not significantly reduce NHO volume in mice [43], implying that IL-1α and IL-1β could have overlapping biological functions as inflammatory mediators in NHO formation. In the initial stages of THO, IL-1β secreted by M1 macrophages was significantly upregulated and participated in the local inflammatory response [9]. Furthermore, in the context of FOP, plasma IL-1β levels were elevated during inflammatory episodes in patients, and anti-IL-1 therapy demonstrated marked therapeutic efficacy in FOP patients [45]. Notably, in the HO microenvironment, IL-1β also acts as an inflammatory amplifier, potentiating the effects of inflammatory mediators including TNF-α, IL-6, and TGF-β1, which synergistically promotes senescence and osteogenic differentiation of TDSC [46]. This highlights the possible involvement of IL-1β in HO formation and suggests that it may be a target for HO therapy.

TNF-α

TNF-α is a pro-inflammatory mediator expressed by M1-type macrophages during HO [9]. Following injury, peripheral blood monocytes migrate to the lesion site, where they differentiate into macrophages and undergo polarization, releasing large quantities of inflammatory cytokines, including TNF-α [14]. These cytokines play a role in establishing an inflammatory microenvironment that promotes osteogenesis in the damaged soft tissue area [14], which in turn influences the osteogenic differentiation of MSCs [28]. TNF-α is likely involved in the early pathological processes of various types of HO. In the early inflammatory phase of THO, TNF-α expression in mouse Achilles tendon tissue rises rapidly following tendon transection, peaks at one week post-injury, and returns to baseline within three weeks [47]. Another study reported significantly elevated serum TNF-α levels in a THO mouse model, with peak levels observed at 48 h post-injury [28]. Similarly, a substantial elevation in serum TNF-α was detected in non-traumatic HO mouse models as well [28].

TNF-α plays a central role in driving the osteogenic differentiation of tendon-derived MSCs. Isaji et al. found that TNF-α downregulates the tendon-specific transcription factor Mohawk (Mkx), thereby accelerating the differentiation of MSCs toward osteogenic and chondrogenic lineages, although the precise regulatory mechanisms remain to be elucidated [47]. Kushima et al. [48] further demonstrated that TNF-α not only participates in the early inflammatory response but also promotes osteoblast differentiation and bone-like tissue formation via activation of the mTOR signaling pathway. In vitro experiments revealed that TNF-α directly induces phosphorylation of mTOR in fibroblasts. In TNF-α knockout mice, mTOR activation was significantly suppressed, and ectopic bone volume was markedly reduced, suggesting that TNF-α is a key upstream regulator of mTOR [48]. Additionally, Hess et al. proposed that TNF-α may mediate MSC osteogenic differentiation by activating the NF-κB signaling pathway [49]. However, the specific role of this mechanism in HO animal models remains to be verified.

#### 3.1.2. Anti-Inflammatory Cytokines

M2 macrophages, recognized as a crucial immunomodulatory cell during the later stages of inflammation, have attracted growing interest for their involvement in modulating MSC function by secreting the anti-inflammatory cytokine IL-10 [29]. Studies have shown that conditioned medium derived from M2-polarized macrophages significantly enhances the osteogenic differentiation potential of MSCs. The neutralization of IL-10 significantly diminishes this effect, as evidenced by reduced expression of osteogenic markers RUNX2 and ALPL, decreased ALP activity and diminished formation of mineralized nodules [29]. Moreover, IL-10 has been found to promote MSC adhesion and migration, further supporting its involvement in bone formation [29]. Although the precise mechanisms through which IL-10 influences the progression of HO remain unclear, existing research has identified a correlation between IL-10 and HO induced by combat-related trauma [50]. Elevated levels of IL-10 have also been observed in the serum or monocytes/macrophages of patients with FOP [51]. Interestingly, in the Nse-BMP4 transgenic mouse model of HO, IL-10 expression remains relatively low during the early stages post-injury, in contrast to the pro-inflammatory cytokines IFN-γ and TNF-α, which are markedly upregulated in the first week and then declined by the fourth week [35]. This inverse temporal pattern suggests that IL-10 could have a more prominent role in the later phase of HO development, possibly serving as a regulatory mechanism that shifts the microenvironment from pro-inflammatory to anti-inflammatory. Consequently, IL-10 has been proposed as a potential biomarker for predicting HO development [14,50]. These findings collectively suggest that IL-10 is a critical effector molecule through which M2 macrophages mediate MSC osteogenic differentiation, potentially contributing to HO by regulating MSC recruitment and osteogenic commitment during ectopic bone formation.

#### 3.1.3. Immune Checkpoint Proteins

Macrophages, as a crucial immune cell population, not only represent a significant proportion in local lesions of HO but also exhibit dynamic fluctuating expression levels of immune checkpoints [35]. Following tissue injury, the expression of ICs shows dynamic alterations during the process of HO: In the early stages, the immune response is overly activated, with a notable upregulation in the expression of stimulatory immune checkpoints (such as CD27, CD28). Meanwhile in the later stages, the immune response is suppressed, and the proportion of cells expressing stimulatory immune checkpoints decreases [35]. At the same time, inhibitory immune checkpoints (such as PD1, TIM3) rise in the early stages, and some inhibitory ICs continue to increase in the later stages [35]. Studies have shown that F4/80+ macrophages broadly express both stimulatory and inhibitory immune checkpoints [35]. The immune response in HO presents a biphasic characteristic, including an early excessive immune response and a later immunosuppressive response. Immune checkpoints are essential in mediating this transition, with macrophages potentially influencing local immune responses through the regulation of immune checkpoint expression.

#### 3.1.4. SIRT1/NF-κB

Sirtuin 1 (SIRT1), a nicotinamide adenine dinucleotide-dependent deacetylase, is essential in modulating inflammation in the context of HO [14,31]. SIRT1 modulates immune responses by deacetylating various target proteins and is recognized as a pivotal immune regulatory factor in the formation of HO [14]. Research has demonstrated that SIRT1 exhibits substantial anti-inflammatory properties, and its activation can effectively slow the progression of HO, particularly by reducing macrophage infiltration and inhibiting the release of pro-inflammatory cytokines such as IL-1β and TNF-α [9,14]. One of the substrates of SIRT1, NFκB p65, is an inflammatory transcription factor that regulates immune responses [14]. In the burn/tenotomy mouse model of HO, immunohistochemistry and Western blot analyses indicated a marked increase in phosphorylated NF-κB p65 levels in the initial phase of the lesion, suggesting a dysregulation of NF-κB signaling in HO [52]. Furthermore, abnormal activation of NF-κB signaling has also been observed in FOP [51]. In HO models, NF-κB signaling mediates the M1 polarization of macrophages and the secretion of associated inflammatory cytokines [14]. Additionally, the hyperactivation of NF-κB signaling in macrophages promotes the release of BMP6 and TGF-β1, which in turn activate the BMP signaling pathway in osteoprogenitor cells, driving the ossification process [52]. The acetylation of NF-κB p65 at the K310 site is crucial for its transcriptional activity. Activation of SIRT1 significantly reduces acetylation at this site, effectively blocking the overactivation of NF-κB signaling and inhibiting heterotopic ossification [14].

#### 3.1.5. Potential Impact of Inflammation on Chondrogenic Differentiation

At the early stages of injury, M1 macrophages infiltrate and release a large amount of pro-inflammatory cytokines, creating an inflammatory microenvironment that promotes the early progression of HO and facilitates the recruitment and proliferation of MSCs [25]. However, as a critical step in the progression of HO, endochondral ossification may be negatively affected by these inflammatory responses. Similar processes are involved in tissue repair through MSC chondrogenic differentiation in osteoarthritis [53]. In osteoarthritis, M1 macrophages directly suppress the expression of cartilage-related genes such as COL2 and ACAN, affecting MSC chondrogenic differentiation [54]. This inhibitory effect is likely related to the cytokines secreted by M1 macrophages [53]. Inflammatory cytokines are destructive to cartilage, inducing chondrocyte senescence and apoptosis, while promoting the release of cartilage-degrading enzymes like MMPs and ADAMTS [55]. A study has demonstrated that in MSC transplantation therapy for cartilage defects, low-intensity pulsed ultrasound stimulation significantly downregulated the expression levels of inflammatory cytokines such as TNF-α and IL-1β, thereby providing a favorable anti-inflammatory microenvironment for newly differentiated chondrocytes and promoting cartilage differentiation and regeneration [56]. In traumatic heterotopic ossification, M2 macrophages, rather than M1 macrophages, enhance MSC chondrogenic differentiation [57]. When MSCs were cultured for 48 h in M1- or M2-conditioned media from the Achilles tendon HO mouse model, we observed a significant increase in the expression of chondrogenic markers such as Sox9 in the M2-conditioned media, while no such effect was seen in the M1-conditioned media [57]. Therefore, we speculate that, compared to M1 macrophages, M2 macrophages in the later stages of HO modulate the immune response by secreting anti-inflammatory and osteogenic factors, thereby providing a favorable anti-inflammatory microenvironment for endochondral ossification.

### 3.2. Macrophage Regulate the Osteogenic Factors and the Osteogenic Pathway

Endochondral ossification is a crucial process in heterotopic ossification, with macrophages playing a key role in regulating chondrogenic and osteogenic differentiation by secreting osteogenic factors, thereby promoting tissue remodeling. We believe that M2 macrophages play a predominant role in this process. In traumatic heterotopic ossification, Tgf-β and VEGF are primarily derived from M2 macrophages, driving endochondral ossification and osteogenic differentiation [57]. MMP9 secreted by M1 and M2c macrophages hydrolyzes the extracellular matrix (ECM), releasing VEGF and Activin A, which promote chondrocyte differentiation and vascular remodeling [30,58,59]. Compared to M1 and M0 macrophages, M2 macrophages secrete BMP-2 at higher levels [60]. Unlike OSM from M1 macrophages, which influences early osteogenic differentiation of MSCs, BMP-2 promotes late-stage osteogenic mineralization in MSCs and plays a critical role in bone remodeling [60]. Additionally, M2-derived extracellular DNA (ecDNA) also contributes to the mineralization of collagen fibers [61]. Therefore, we discuss here the osteogenic factors derived from macrophages and their underlying pathogenic mechanisms.

#### 3.2.1. Activin A

Activin A, a member of the TGF-β superfamily, is secreted by activated macrophages, which are likely the main source of Activin A in human inflammatory response [30,33]. Studies have shown that Activin A accelerates the progression of acquired HO and plays a crucial role in the onset and progression of FOP [6,30,62,63]. Recent studies suggest that Activin A in FOP may primarily originate from M1 macrophages, working synergistically with TGF-β and other inflammatory cytokines to contribute to the progression of HO [30,33]. However, Matsuo et al. proposed that Activin A does not affect the activation state of macrophages or the release of pro-inflammatory cytokines and does not involve a feedback regulation mechanism. This suggests that the increase in Activin A is due to excessive macrophage activation rather than being regulated by its own levels [33].

The mechanism by which Activin A induces HO involves the complex regulation of multiple signaling pathways, further driving the development of HO. Firstly, both Activin A and TGF-β can influence the progression of HO through the pSMAD2/3 signaling pathway [30]. Activin A initiates signaling by binding to type II receptors (ACVR2A, ACVR2B, and BMPR2) and type I receptors (ACVR1B/1C), which activates the downstream pSMAD2/3 signaling pathway, promoting the expression of chondrogenic genes such as Sox9, Acan, and Col2a1, and driving chondrogenic cell differentiation (Figure 3) [6,30]. This pathway may act upstream of the BMP signaling pathway, and inhibitors of pSMAD2/3 signaling notably suppress chondrogenesis induced by BMP [30,64]. In FOP, Activin A can form a complex with both type I (ACVR1) and type II receptors [6]. When Activin A binds to wild-type ACVR1, it does not trigger signaling pathways but forms non-signaling complexes, blocking the normal BMP signaling mediated by ACVR1 and limiting ectopic bone formation [63]. However, mutations in ACVR1 lead to abnormal responses to Activin A [63]. When Activin A interacts with the ACVR1 receptor R206H variant in FOP, it activates the typically BMP-induced pSMAD1/5/8 signaling pathway, leading to abnormal bone formation [6,65]. Furthermore, in some cases, Activin A can antagonize BMP6 signaling; however, in HO, Activin A enhances the pro-HO effects of BMP6 in vivo without affecting the BMP6-induced cartilage formation in vitro [6,30]. This suggests that the function of Activin A depends on the specific biological context and the particular combination of receptors and ligands in different cell types [30].

In addition to macrophages and other activated immune cells, significant expression of Inhba (the gene encoding Activin A) has also been found in HO progenitor cells, overlapping with the expression of chondrogenic genes. This could promote ectopic bone growth and progenitor cell differentiation into cartilage. Macrophages and mast cells may regulate the expression of Inhba in HO progenitor cells through the secretion of inflammatory cytokines, such as TNF and IL-1β [30].

#### 3.2.2. Matrix Metalloproteinase

The matrix metalloproteinase (MMP) family consists of over 20 enzymes that are essential for bone formation and healing by remodeling the extracellular matrix (ECM) and releasing bioactive molecules [66]. MMP-9, an enzyme secreted by several cell types such as neutrophils, fibroblasts, and macrophages, is recognized as a potential diagnostic biomarker for early HO [67]. In trauma-induced FOP flare-ups, activated pro-inflammatory macrophages secrete MMP-9, which becomes a key regulator in the progression of HO following inflammation and tissue damage [59]. As a factor involved in angiogenesis during endochondral ossification, MMP-9 is secreted at higher levels by M2c macrophages, whereas M2a macrophages secrete significantly less MMP-9 compared to other macrophage phenotypes [58,68].

Macrophage-derived MMP-9 degrades the protein components of the ECM through its proteolytic activity, particularly targeting heparan sulfate proteoglycans (HSPGs) [59,69]. This degradation not only alters the structure of the ECM but also releases bioactive molecules that were bound to HSPGs in the ECM, such as VEGF and Activin A, which play critical roles in endochondral ossification and vascular remodeling during HO [58,59]. Angiogenesis and vascular infiltration promote hypertrophic chondrocyte apoptosis and MSC recruitment, thereby creating a favorable environment for endochondral ossification [10]. Studies have shown that M2 macrophages accumulate during traumatic HO and produce large amounts of VEGF, which promotes angiogenesis and vascular development (Figure 3) [57].

Moreover, MMP-9 shapes the microenvironment by modifying the ECM’s rigidity and composition, creating a favorable environment for HO formation. This microenvironment provides a foundation for the misinterpretation of signals by FAPs carrying the ACVR1R206H mutation, which incorrectly initiates chondrogenesis and leads to HO [59,70,71]. Therefore, the role of MMP-9 in FOP pathogenesis is complex, as macrophage-secreted MMP-9 may influence the progression of HO through various mechanisms, including degradation and remodeling of the ECM and release of bioactive molecules (Figure 3) [59].

Additionally, MMP-10 (Stromelysin 2) has been demonstrated to contribute to the formation of FOP [72]. A study on mouse fracture healing found that during intramembranous and endochondral ossification, MMP-10 secreted by macrophages/monocytes activates proMMP-9 secreted by these cells, promoting vascularization of the cartilaginous callus and thus enhancing fracture healing [73]. This finding may provide insights into the mechanisms by which macrophage-secreted MMPs influence heterotopic ossification.

#### 3.2.3. Tgf-β Signaling Pathway

TGF-β1, an osteogenic cytokine, is secreted by M2 anti-inflammatory macrophages [33] and activated through ECM or LAP degradation induced by inflammation [64]. Its signaling is upregulated at the HO anlagen, stimulating chondrogenesis in mesenchymal stem cells and inducing endochondral ossification [15,74,75]. Activation of TGF-β1 triggers the heterodimerization of TGF-βRII and TGF-βRI, initiating the classical TGF-β signaling pathway. In this process, SMAD2/3 is phosphorylated and translocated to the nucleus, where it activates the transcription of related genes (Figure 3) [75]. TGF-β signaling in the pathophysiology of HO is complex, especially regarding macrophages’ role in the expression and regulation of TGF-β ligands and receptors. During injury, TGF-β1 acts through autocrine secretion by macrophages [75]. Elevated TGF-β1 at the wound site attracts inflammatory cells, including macrophages [76], which then move to the lesion area and release cytokines, with TGF-β1 being released in elevated amounts [75].

scRNA-Seq analysis revealed that compared to uninjured controls, 7 days after burn/tenotomy HO modeling in mice, the expression of several TGF-β-stimulated genes was upregulated in both macrophages and mesenchymal stem cells. These genes are crucial for immune function (such as Apoe, Cebpb, and Fn1) and for synthesis, adhesion to, or restructuring of the extracellular matrix (such as Fn1, Col1a1, and Col3a1) [75]. Furthermore, compared to mesenchymal stem cells, the expression of Tgfb1 and its receptors Tgfbr1 and Tgfbr2 in macrophages was significantly elevated over time in injured mice [75]. snATAC-Seq analysis on macrophage chromatin accessibility of Tgfb receptor genes confirmed this finding, showing that macrophages exhibit increased sensitivity to TGF-β signaling [75]. This suggests that TGF-β signaling plays a critical role in the progression of HO in both macrophages and mesenchymal progenitor cells (MPCs), particularly as TGF-β autocrine signaling from infiltrating macrophages promotes early post-injury HO formation. In addition to traumatic heterotopic ossification, activated TGF-β signaling is also a common pathological mechanism in the progression of HO in FOP [64]. However, macrophage-mediated TGF-β signaling has not been observed in the process of intramembranous ossification in POH [64].

#### 3.2.4. BMP

BMPs, members of the TGF-β superfamily, act as osteogenic factors that are crucial for the regulation of bone homeostasis [77]. Aberrant activation of BMP-2 is closely associated with the development of HO [78]. Secreted by M2 macrophages, BMP-2 has been shown to induce chondrogenesis and serves as a key factor in promoting osteoblast differentiation during bone healing [23,78]. BMP-2 exerts its effects by phosphorylating SMAD1/5/9, which then forms a complex with SMAD4. This R-SMAD-SMAD4 complex translocates into the nucleus, where it promotes the transcription of genes involved in chondrocyte and osteoblast differentiation as well as bone formation [79]. Recent research indicates that Discoidin Domain Receptor 2 (DDR2) plays a modulatory role in BMP-2-induced HO by interacting with collagen and facilitating the nuclear accumulation of YAP/TAZ, thereby amplifying BMP-2 signaling [79]. Rather than directly targeting the canonical SMAD pathway, DDR2 exerts its influence by suppressing LATS1/2 activity, which prevents YAP/TAZ degradation via the proteasome [79]. The enhanced nuclear presence of YAP/TAZ subsequently boosts the transcriptional function of the R-SMAD-SMAD4 complex, promoting the activation of osteogenesis-related genes (Figure 3) [79].

Co-culture of preosteoblasts and macrophages showed that after lipopolysaccharide (LPS) stimulation, macrophage secretion of BMP-6 increased under inflammatory conditions, leading to abnormal activation of the BMP pathway in preosteoblasts, thereby promoting osteoblast differentiation [52]. Recent research has revealed dysregulation of the BMP pathway in both acquired HO and FOP [80]. When BMP binds to its type I and type II receptors, the resulting complex further phosphorylates downstream Smad1/5/8 proteins [81]. In ectopic bone formation caused by muscle injury, BMP-7 is significantly upregulated in the early stages and co-localizes with macrophages. In vivo and in vitro inhibition of BMP-7 reduces osteogenesis and ectopic bone formation, confirming BMP-7′s role in enhancing abnormal osteogenic signaling in traumatic heterotopic ossification [82]. Under the pathological condition of FOP, BMP signaling induces macrophage activation and inflammatory cytokine secretion, leading to abnormal inflammatory responses. Furthermore, in the early lesions of FOP, BMP signaling stimulates the upregulation of substance P secretion, which induces mast cell degranulation, promoting osteoblast migration and ultimately leading to ectopic bone formation [83]. However, reports on the potential role of BMPs in the pathogenesis of NHO are limited and remain to be proven [84].

#### 3.2.5. Macrophage-Derived Extracellular DNA

In recent years, ecDNA has been identified as an important molecule in various pathological environments, carrying multiple anions, which allows it to effectively regulate the stability of calcium and phosphate [61]. EcDNA is involved in ectopic calcification processes in tissues such as gallstones and certain cancers, showing great potential in triggering biomineralization [85,86,87].

In a model of Achilles tendon injury in rats, ectopic calcification and ecDNA content both markedly increased with the progression of HO. Deoxyribonuclease degradation of ecDNA prevented ectopic bone formation and in vitro calcification in the rat model, further demonstrating the role of ecDNA in ectopic calcification [61]. Co-culturing with M2 macrophages led to significant mineralization of type I collagen. EcDNA released from M2 macrophages, enriched at the injury site, was transported to the ectopic ossification area, promoting collagen fibril mineralization and thus accelerating HO progression (Figure 3) [61]. However, the study lacked in-depth investigation into the pattern of ecDNA release by macrophages, and whether ecDNA is released after M2 macrophage death or through other means such as extracellular vesicles require further exploration.

## 4. HO Treatment Target Macrophage

Traditional treatments for HO mainly focus on suppressing inflammation and delaying bone formation. Common therapeutic approaches include surgical excision, nonsteroidal anti-inflammatory drugs (NSAIDs), and radiation therapy [1,84,88]. However, these methods have several limitations, such as a high recurrence rate after surgery [1], gastrointestinal complications caused by NSAIDs [89], and the potential for sepsis due to radiation therapy [90].

New therapeutic approaches must be developed in light of a better understanding of the pathophysiology of HO, including the involvement of different subtypes of macrophages play in its development. According to recent studies, macrophages of different polarization states may be crucial for osteogenesis and the inflammatory response during the development of HO. This indicates that targeting the macrophage-driven immune microenvironment and regulating the downstream osteogenic pathways of macrophages could offer new directions for HO treatment.

### 4.1. Drugs That Modulate the Macrophage and Inflammatory Microenvironment

#### 4.1.1. Interleukin-1 Inhibitors

IL-1β, primarily secreted by M1 macrophages, serves as a crucial inflammatory mediator in driving osteogenesis and advancing the progression of HO [42,46]. Multiple clinical cases have confirmed that IL-1β levels significantly increase during HO progression, and treatment with anti-IL-1β drugs can improve symptoms and progression in patients with FOP [45]. Commonly used IL-1 inhibitors, such as anakinra (IL-1 receptor antagonist) and canakinumab (anti-IL-1β monoclonal antibody), are already established in the treatment of various other inflammatory conditions [91]. In pediatric patients with FOP, IL-1 inhibitors have demonstrated efficacy in alleviating pain and inflammation associated with HO, reducing the frequency of flare-ups, and minimizing the size of ectopic bone growths. These therapies have shown promising long-term tolerability, with minimal adverse effects, making them a promising treatment option for FOP [92]. However, the effectiveness of IL-1 inhibitors in different patient types and the optimal dosing regimen remain unclear and require further research for validation [92].

#### 4.1.2. Palovarotene

Palovarotene is a synthetic retinoic acid receptor γ (RARγ) agonist that has been approved by Health Canada and the U.S. Food and Drug Administration (FDA) for the treatment of FOP [92]. Research indicates that Palovarotene may inhibit the formation and progression of HO through multiple mechanisms. The inflammatory microenvironment of HO drives osteogenic differentiation of tendon stem cells (TSCs), and Palovarotene modulates the HO immune microenvironment, predominantly through macrophage. It alters the inflammatory features at the injury site, inhibiting the osteogenic differentiation tendency of TSCs [26]. In a combat-related HO rat model, Palovarotene treatment reduced the infiltration of inflammatory cells at the injury site [93]. Compared to the control group, the Palovarotene-treated group showed reduced macrophage recruitment, effectively alleviating the development of Achilles tendon HO [26]. Additionally, Palovarotene treatment significantly reduced systemic inflammation and lowered the secretion of pro-inflammatory cytokines, such as IL-1β, IL-6, and IFN-γ, as well as TNF-α secreted by macrophages, demonstrating strong anti-inflammatory effects (Figure 4) [93].

Interestingly, there is evidence suggesting that the therapeutic effect of Palovarotene on HO may be related to its regulation of Activin A [94]. After Palovarotene administration, the number of chondrogenic cells and macrophages expressing the Activin A-encoding gene Inhba was significantly reduced, weakening the role of Activin A in the HO microenvironment [94]. However, Palovarotene did not directly inhibit the expression of Inhba [94]. These results provide additional insight into the mechanisms of Palovarotene in treating HO.

Palovarotene has been approved by Health Canada and the U.S. Food and Drug Administration for the treatment of HO in females aged 8 and older as well as males aged 10 and older who are diagnosed with FOP [92]. However, management of acute flare-ups and new sites of HO continues to be inadequate.

#### 4.1.3. Metformin

Metformin is an FDA-approved medication for type 2 diabetes [95], and has potential therapeutic benefits for various inflammation-related diseases, including ulcerative colitis [96] and atherosclerosis [97]. Metformin could regulate immune cell differentiation and exert anti-inflammatory effects, particularly influencing macrophage differentiation and the inflammatory responses they mediate [52]. The transformation of monocytes into macrophages is a critical step in mediating inflammatory responses. Metformin not only inhibits this differentiation process but also reduces the secretion of pro-inflammatory cytokines [52,98]. By modulating the AMPK pathway, metformin reduces pro-inflammatory M1 macrophages and induces polarization toward the anti-inflammatory M2 macrophage phenotype, demonstrating strong anti-inflammatory effects [52].

Metformin also prevents post-injury cartilage formation and the progression of HO by modulating macrophage-driven inflammation [31]. In both in vitro and in vivo experiments, metformin treatment reduced the expression of macrophage marker F4/80 in the lesion area and dose-dependently decreased the release of inflammatory cytokines such as IL-1β, TNF-α, and MCP-1, thereby exerting its anti-inflammatory effects [31]. Mechanistically, metformin increases the expression of SIRT1 in a dose-dependent manner. SIRT1 deacetylates the downstream pro-inflammatory mediator NF-κB p65, which reduces inflammation in the HO microenvironment and regulates cytokine transcription (Figure 4) [31]. By activating AMPK, metformin weakens macrophage infiltration in the immune microenvironment, preventing and inhibiting the progression of HO [52].

#### 4.1.4. Parishin A-Loaded Mesoporous Silica Nanoparticles

Parishin A (PA) is a natural small molecule compound extracted from the traditional Chinese medicine Gastrodia elata, which exhibits significant anti-inflammatory effects and effectively reduces HO induced by tendon injury [32]. PA targets the polarization of M2 macrophages, effectively alleviating the progression of HO. The JAK-STAT signaling pathway is one of the core mechanisms by which PA treats heterotopic ossification. This pathway is crucial for regulating macrophage function and mediating macrophage polarization, and it plays a role in the progression of various inflammation-related diseases [32,99]. PA promotes the polarization of macrophages toward the anti-inflammatory M2 state by inhibiting STAT1 gene transcription and STAT1 phosphorylation at the S727 site, thereby modulating and suppressing the inflammatory response at the site of the injured tendon (Figure 4) [32]. This regulatory effect also effectively inhibits the expression of pro-inflammatory cytokines such as IL-6, IL-1β, and TNF-α, improving the paracrine function of macrophages in the immune microenvironment [32]. Under the influence of PA, an anti-inflammatory environment is established for the TSPCs at the injured tendon site, which in turn inhibits the osteogenic and chondrogenic differentiation of TSPCs, contributing to further suppression of heterotopic ossification in the tendon [32].

Compared to PA solution injections, nanoparticle-based delivery systems, such as MSN@PA, achieve sustained release of PA at the tendon injury site, and animal experiments have shown faster tendon repair and fewer injection frequencies [32,100]. However, this therapy has not yet undergone clinical studies and still requires evaluation in large animal models [32].

#### 4.1.5. Matrine

Matrine, an active compound extracted from Chinese herbs such as Sophora alopecuroides, possesses anti-inflammatory, antihypertensive, and antitumor properties [101]. Previous studies have identified its potential in treating HO by interfering with the activation of the TGF-β/Smad2/3 signaling pathway, thereby inhibiting the migration and osteogenic differentiation of MSCs to halt HO progression [102]. Recent research has further clarified Matrine’s mechanisms in HO treatment, emphasizing its role in modulating macrophage polarization.

In the early HO microenvironment, M2 macrophages secrete various anti-inflammatory and osteogenic cytokines, including TGF-β and VEGF, which promote the osteogenic differentiation of precursor cells [57,103]. Matrine targets the MAPK signaling pathway, a crucial modulator of macrophage function and polarization in bone marrow-derived macrophages (BMDMs). It suppresses the phosphorylation of key MAPK components ERK and P38, which are associated with the pro-inflammatory activities of macrophages, thereby preventing M2 macrophage polarization (Figure 4) [12]. By targeting macrophage polarization, Matrine indirectly inhibits the osteogenic differentiation of progenitor cells and the progression of trauma-induced HO [12].

#### 4.1.6. Spray of Hydrogel

Curcumin, a compound isolated from Curcuma, has been widely applied in treating a range of diseases, including ulcerative colitis [104] and arthritis [105], as well as demonstrating anticancer biological activities [106]. It also contributes to the regulation of M1/M2 macrophage polarization [104]. The Cur@ZIF-8@CeO2 (CZC) nano-metal–organic framework drug delivery system incorporates curcumin into a multi-layered nanostructure. Using HAMA hydrogel spray technology, the system forms a hydrogel membrane to cover the trauma site. This system gradually releases curcumin in the acidic inflammatory microenvironment, regulating immune homeostasis and macrophage polarization at the trauma site while enhancing stem cell homeostasis. Through this dual homeostasis modulation, it aims to prevent and manage early THO [107].

After CZCH treatment, macrophage functional homeostasis in the early inflammatory microenvironment of THO was significantly improved. Curcumin effectively suppressed excessive inflammatory responses caused by abnormal immune cell infiltration. The M1/M2 macrophage ratio normalized, and in inflammation-activated macrophages, NF-κB, IL-17, and mTOR signaling pathways were inhibited. The secretion of pro-inflammatory cytokines and chemokines such as IL-6 and TNF-α were downregulated (Figure 4) [107]. Additionally, macrophage exocytosis was enhanced, further promoting the recovery and maintenance of immune homeostasis [107]. Animal model studies demonstrated that CZCH treatment effectively suppressed early inflammatory responses in THO mice. Unlike the control group, the CZCH-treated mice did not exhibit significant heterotopic bone formation during the early inflammatory phase. Furthermore, CZCH displayed good biocompatibility, highlighting its broad potential for clinical applications [107].

#### 4.1.7. Fetuin-A

As a cystatin superfamily protease inhibitor, Fetuin-A is a well-recognized inhibitor of ectopic calcification [108,109] and has shown therapeutic and diagnostic potential in various diseases, including calcification in osteoarthritis [110], psoriasis [111] and coronary artery calcification [112]. In recent years, Fetuin-A has been implicated in HO. Its expression is significantly downregulated in SCI-mediated NHO [113] and Achilles tendon injury-induced HO [114]. Recent studies have further demonstrated that Fetuin-A promotes M2 macrophage polarization and stimulates the PD-1 pathway to induce immunosuppression, suggesting its potential as a therapeutic approach for HO [24].

During the early inflammatory phase of HO, M1 macrophages secrete pro-inflammatory cytokines that drive cell proliferation [51]. At around 3 dpi, anti-inflammatory macrophages M2, derived from M1, promoted chondrogenic differentiation of MSCs [25]. Leveraging Fetuin-A to promote the early transition of M1 to M2 macrophages can effectively modulate immune responses, suppress inflammation, support muscle repair [115], and control HO progression [24] (Figure 4). Immune checkpoints, which influence HO progression and regulate the immunosuppressive microenvironment, also play a crucial role in Fetuin-A-induced M2 polarization [24,35]. As an immunomodulator, Fetuin-A can reduce macrophage infiltration or enhance macrophage polarization and osteoclast activation by stimulating IC molecules [24]. Notably, Fetuin-A-induced upregulation of PD-1 exhibits strong anti-inflammatory effects, effectively suppressing excessive immune responses [24]. Targeting Fetuin-A may represent a promising avenue for exploring novel HO therapies. By regulating the function of Fetuin-A, it may be possible to develop innovative treatment strategies to address HO and associated bone loss issues.

#### 4.1.8. Quercetin

Quercetin, a flavonoid-class polyphenol derived from natural herbs, is known to regulate immune and inflammatory responses in various diseases [116]. In traumatic HO, quercetin alleviates excessive inflammation at the injury site by modulating the SIRT1/NFκB pathway, inhibiting monocyte-to-macrophage transition and macrophage polarization, making it a promising therapeutic candidate for HO [14]. Following tendon injury, the expression of SIRT1 protein decreases, accompanied by a reduction in downstream acetylated NFκB p65. This signaling pathway is involved in monocyte conversion and M1 macrophage polarization. Quercetin reverses this mechanism by upregulating SIRT1 expression, thereby suppressing macrophage infiltration and migration in a dose-dependent manner and restoring immune homeostasis (Figure 4) [14]. In addition to regulating macrophage polarization, quercetin negatively impacts the secretion of inflammatory cytokines such as TNF-α, IL-1β, and IL-6 produced by macrophages, further preventing and controlling trauma-induced inflammation and HO progression [14]. With a high safety profile and minimal reported side effects in current studies, quercetin demonstrates significant potential for clinical application [117].

#### 4.1.9. CD47-Activating Peptides

In the treatment of THO, the CD47-activating peptide (p7N3) modulates monocyte and macrophage phenotypes, accompanied by a downregulation in the secretion of the osteogenic factor TGF-β1 [15]. In a burn/tendon transection mouse model, p7N3 treatment significantly reduced cartilage and mature HO formation. Changes in macrophage phenotype and function were observed, with decreased expression of M2 macrophage markers (Arg1 and Mrc1) and increased expression of the M1 macrophage marker (iNos), indicating a shift from a reparative to a pro-inflammatory macrophage phenotype (Figure 4). However, the specific mechanisms by which p7N3 induces macrophage polarization remain unclear [15]. In the early phases of inflammation, macrophages release TGF-β1, which contributes to wound healing, inflammatory reactions, and the abnormal development of mesenchymal stem cells. According to earlier research, CD47 regulates the TGF-β signaling pathway [118]. Nevertheless, CD47-activating peptide treatment did not significantly reduce TGF-β1 levels at the wound site or in serum, indicating that the inhibitory effect of the CD47-activating peptide on HO is primarily due to its regulation of macrophage phenotypes [15].

#### 4.1.10. Focal Adhesion Kinase-2 Inhibitor

One essential tyrosine kinase that controls cell adherence to the extracellular matrix is focal adhesion kinase-2 (FAK2) [119]. It facilitates cellular migration, proliferation, and survival by activating downstream signaling pathways, such as PI3K-AKT1 and MAPK1/ERK2 [120]. Increasing evidence suggests that FAK plays a central role in mediating various pathological conditions by modulating macrophage proliferation, migration, pro-inflammatory activity, and signal transduction [121,122,123]. A recent study showed that treatment with a FAK2 inhibitor (PF-573228) led to comparatively higher expression of M2 macrophage markers (e.g., Arg1, CD163, Mrc1, Ccl2, Tgfb1) and lower expression of M1 markers (e.g., Nos2, Cxcl2, Il1a, Cd80) in a rat model of blast-associated complex lower limb injury (Figure 4). Furthermore, the FAK2 inhibitor group exhibited reduced production of pro-inflammatory cytokines, including IL-6 [119]. These findings emphasize that FAK2 inhibitors can facilitate the polarization of macrophages from the M1 to the M2 phenotype, thereby alleviating early inflammation, accelerating tissue repair, reducing osteogenic signaling, and ultimately inhibiting heterotopic ossification (HO) formation in trauma-induced HO treatment [119].

#### 4.1.11. Ethyl Caffeate (ECF)

The traditional Chinese herb Artemisia aucheri [124] contains an active component called ECF, which has no direct effect on MSC osteogenic development. Rather, by controlling inflammatory signaling pathways and macrophage polarization, it prevents the advancement of HO [9]. According to Wang et al., ECF decreases the synthesis of pro-inflammatory cytokines and suppresses macrophage polarization toward the M1 phenotype in a dose-dependent way [9]. ECF specifically targets and activates SIRT1, which subsequently deacetylates and inhibits NF-κB activity (Figure 4). In M1 macrophages, this inhibition lowers the expression of inflammatory cytokines such TNF-α and IL-6. Consequently, the expression of osteogenic genes, including ALP, Runx2, and OCN, is downregulated, thereby suppressing MSC osteogenic differentiation [9].

### 4.2. Drugs That Affect Osteogenic Signaling in Macrophages

#### 4.2.1. TGF-βRII-Fc

During endochondral ossification, TGF-β1 is expressed by early-infiltrating M2 macrophages and functions in an autocrine manner [75]. It regulates the recruitment of circulating inflammatory immune cells to the injury site [76], facilitates the recruitment and activation of fibroblasts [125,126], and stimulates chondrogenesis in MPCs [15]. Activation of TGF-β1 induces the formation of a heterodimeric receptor complex comprising TGF-βRII and TGF-βRI, initiating the canonical downstream TGF-β signaling pathway [127]. Since the absence of the TGF-βRI receptor signaling alone has limited effects on HO progression, this study bypassed targeting the receptor signaling. Instead, it utilized a ligand trap, TGF-βRII-Fc, to target the upstream TGF-β1 ligand signaling of TGF-βRI (Figure 5). This approach showed potential as an effective treatment for preventing and alleviating HO progression, resulting in favorable outcomes in HO mouse models [75].

#### 4.2.2. Galunisertib

As a TGF-β receptor I (TβRI) kinase inhibitor, Galunisertib (Gst) has potential therapeutic effects on traumatic HO [128]. In a mouse model of an Achilles tendon puncture-induced ATP HO, Gst treatment demonstrated inhibitory effects on the inflammation phase and endochondral ossification during the cartilage formation phase of HO [128]. The inhibitory effect was dose-dependent, with increasing doses leading to stronger suppression. Immunofluorescence and Western blot analysis revealed that Gst treatment significantly downregulated phosphorylated TGF-β receptor I (p-TβRI) and its downstream signaling molecules, p-Smad2/3, in TDSCs (Figure 5) [128]. This suggests that Gst inhibits the phosphorylation of TGF-β receptor I, affecting the downstream Smad2/3 signaling pathway and the transcription of osteogenic genes, thereby effectively suppressing the progression of traumatic HO [128].

#### 4.2.3. Ruxolitinib

OSM acts as a risk factor in the development of NHO [38]. After binding to the OSMR expressed on muscle satellite and interstitial cells, OSM activates the JAK1/2 tyrosine kinase, leading to the tyrosine phosphorylation of the downstream transcription factor STAT3 [41]. Ruxolitinib, a JAK1/2 tyrosine kinase inhibitor, significantly reduces STAT3 activation in injured muscles by inhibiting JAK1/2 (Figure 5) [41]. Animal studies have shown that after Ruxolitinib treatment, the phosphorylation of STAT3 and the volume of SCI-induced NHO lesions were significantly reduced, suggesting that the JAK/STAT3 signaling pathway could be an effective target for NHO therapy [41].

#### 4.2.4. Garetosmab

ACVR1R206H is the pathogenic cause of FOP. Activin A, secreted by macrophages, binds to the mutated ACVR1 receptor, driving the activation of FOP [63]. Studies have confirmed that the ACVR1R206H mutation does not result in constitutive activity but requires ligand activation, particularly from BMP/TGF-β family ligands [63,65]. This has made the inhibition of ACVR1 ligands a promising drug target for FOP treatment [63]. Garetosmab is a monoclonal antibody against Activin A that binds to Activin A and blocks its interaction with the mutated ACVR1 receptor, and it is under investigation for the treatment of FOP (Figure 5).

In phase 1 clinical trials (ClinicalTrials.gov identifier NCT02870400, https://clinicaltrials.gov/study/NCT02870400, accessed on 1 April 2025), Garetosmab showed good tolerability and acceptable safety. After intravenous administration of a 10 mg/kg dose, the total Activin A levels in the blood remained stable for 4 to 12 weeks, supporting further investigation of the drug for FOP treatment [129]. The phase 2 LUMINA-1 trial (ClinicalTrials.gov identifier NCT03188666, https://clinicaltrials.gov/study/NCT03188666, accessed on 1 April 2025) evaluated the safety and efficacy of Garetosmab in the treatment of HO. The trial included 44 FOP patients aged 18 to 60 years in a double-blind, placebo-controlled study. Compared to the placebo group, the Garetosmab group had a higher incidence of mild to moderate adverse events, such as epistaxis and madarosis. However, after the first treatment period, there were fewer new HO lesions and a lower frequency of FOP flare-ups. In period 2, the placebo group treated with Garetosmab showed a 95% reduction in HO lesions [130]. Based on these results, Garetosmab is currently undergoing phase 3 trials in adult FOP patients (OPTIMA, NCT05394116) [63].

#### 4.2.5. Rapamycin

mTOR signaling plays a crucial role in processes such as cartilage formation and osteogenesis [131,132]. Rapamycin, an mTOR inhibitor, suppresses the proliferation and differentiation of osteoblasts, showing therapeutic potential for both traumatic and genetic heterotopic ossification [131,133]. In traumatic heterotopic ossification, mTOR is activated by the upstream inflammatory factor TNF-α [48]. In an Achilles tendon tenotomy mouse model, continuous administration of rapamycin significantly reduced mTOR phosphorylation levels at the tenotomy site and, compared to the control group, decreased the volume of ectopic bone formation [48]. In FOP, mTOR signaling is abnormally activated through the Activin A/FOP-ACVR1/ENPP2 cascade, promoting cartilage formation and heterotopic ossification. Rapamycin, by inhibiting the mTOR pathway, can significantly reduce or prevent the formation of FOP (Figure 5) [131].

## 5. Conclusions

Heterotopic ossification HO is a significant clinical challenge. Recent studies have revealed the complex role of macrophages in HO, highlighting their dual functions in inflammation and tissue repair. Different macrophage subtypes dominate at various stages of HO progression: M1 macrophages play a key role in triggering the inflammatory microenvironment, promoting MSCs proliferation and chondrogenic differentiation, while M2 macrophages are primarily involved in immune suppression within the HO microenvironment and facilitate osteogenic and chondrogenic differentiation of MSCs. However, the dynamic polarization of macrophages and their pathogenic mechanisms in the development of HO require further investigation.

Specifically, macrophages play an active role in the initiation and progression of HO by modulating the inflammatory microenvironment and influencing osteogenic factors and pathways. Within the inflammatory microenvironment, macrophages are influenced by inflammation regulators and secrete pro-inflammatory and anti-inflammatory cytokines, as well as immune checkpoint proteins, thereby maintaining the balance of the immune environment and regulating MSC osteogenic and chondrogenic differentiation, ultimately affecting ectopic bone formation. Additionally, macrophages activate osteogenic factors such as activin A, TGF-β, and BMP, targeting MSCs and other HO-initiating cells, thereby triggering downstream osteogenic pathways and driving the heterotopic ossification process.

As our understanding of macrophages involvement in HO deepens, innovative therapies targeting macrophage functions hold promise for improving patient outcomes. Emerging interventions, such as immunomodulators, osteogenic pathway inhibitors, and drugs that regulate macrophage polarization, have shown promising results in preclinical studies. The continued development of macrophage-based, individualized treatment strategies may open new avenues for the effective prevention and treatment of HO.

In the future, novel technologies such as single-cell and spatial transcriptomics will help further explore the various macrophage subtypes in HO, their dynamic changes at different stages, and how they promote HO progression. This will deepen our understanding of the immune microenvironment in HO and provide new insights into the interactions between macrophages and other immune cells in the disease process. Developing new drugs targeting macrophage polarization and function is expected to become a key direction in HO treatment. Given the complex pathological process of HO, multi-target combination therapies may prove to be more effective. Finally, the integration of preclinical models and clinical trials will help validate the efficacy and safety of these novel therapeutic strategies, offering more precise treatment options for HO patients.

## Figures and Tables

**Figure 1 ijms-26-05821-f001:**
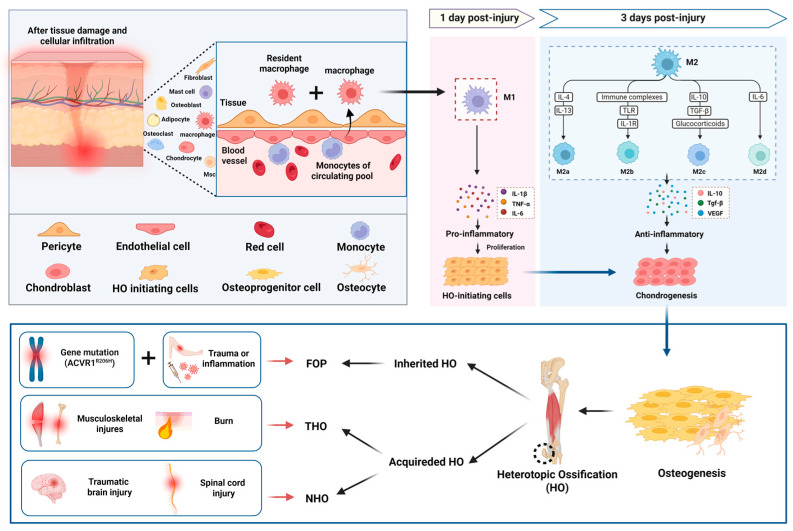
The role of macrophage polarization in the pathogenesis of ossification. HO can be hereditary, as in FOP, or acquired, often following musculoskeletal trauma, burns, or CNS injuries. In both cases, tissue damage triggers the recruitment of macrophages, including resident macrophages and monocyte-derived macrophages. M1 macrophages infiltrate within 1 day post-injury, releasing pro-inflammatory cytokines (e.g., TNF-α, IL-β, IL-6) that stimulate the proliferation of HO-initiating cells and inflammatory responses. By 3 days post-injury, M2 macrophages (M2a, M2b, M2c, M2d) become dominant, secreting anti-inflammatory and growth factors (e.g., IL-10, TGF-β, VEGF) that promote chondrogenesis, osteogenesis, and tissue remodeling. Thus, the polarization of macrophages from M1 to M2 plays a critical role in regulating HO progression.

**Figure 2 ijms-26-05821-f002:**
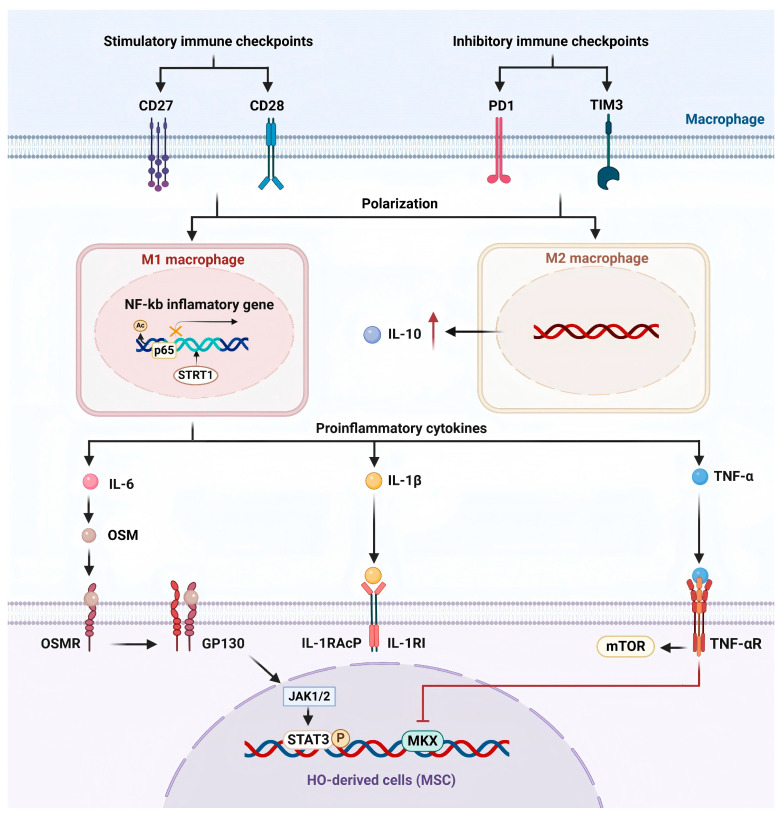
Mechanism diagram of macrophages promoting HO formation by regulating the inflammatory microenvironment. Dysregulation of SIRT1/NF-κB in macrophages affects the transcription of inflammatory cytokines in HO. During the early inflammatory phase of HO, M1 macrophages activate the local inflammatory response by secreting pro-inflammatory cytokines (such as TNF-α, IL-1β, and IL-6). In the later stage of inflammation, anti-inflammatory cytokines, including IL-10, are secreted, which inhibit the inflammatory response. Dynamic changes in macrophage immune checkpoint expression are involved in the HO immune microenvironment.

**Figure 3 ijms-26-05821-f003:**
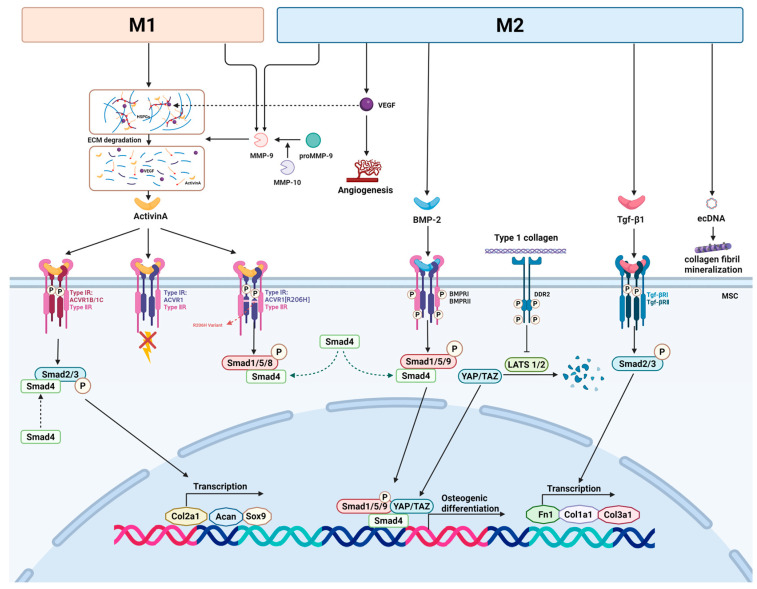
Role of Macrophages in Regulating Osteogenic Factors and Pathways in Heterotopic Ossification. Macrophages secrete several factors that activate key signaling pathways involved in HO progression. Activin A, secreted by M1 macrophages, binds to its receptors, triggering the SMAD2/3 pathway, which promotes chondrogenic differentiation. When Activin A binds to the ACVR1 R206H variant in FOP, it enhances the SMAD1/5/8 pathway, further driving abnormal bone formation. MMP9 derived from M2 and M1 macrophages modulates the ECM structure, releasing bioactive molecules such as Activin A and VEGF, thereby influencing the progression of HO. M2 macrophages secrete TGF-β1, which activates the SMAD2/3 pathway, stimulating the nuclear transcription of osteogenic factors. BMP-2 derived from M2 macrophages phosphorylates SMAD1/5/9 and binds to SMAD4, promoting transcription of genes favoring chondrogenic and osteogenic differentiation. M2 macrophage-derived ecDNA further promotes mineralization and accelerates the progression of HO.

**Figure 4 ijms-26-05821-f004:**
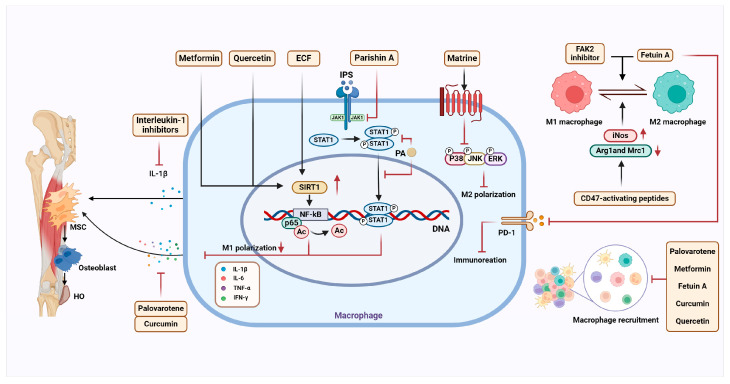
Mechanisms of pharmacological modulation of macrophages and inflammatory microenvironment in the treatment of HO. The figure includes different drugs (e.g., IL-1 inhibitors, Palovarotene, metformin, Parishin A carrier, etc.) that improve the immune microenvironment by modulating macrophage polarization, inhibiting inflammatory responses, and modulating the levels of relevant cytokines (e.g., IL-1β, TNF-α, IL-6, etc.), thereby inhibiting the progression of HO.

**Figure 5 ijms-26-05821-f005:**
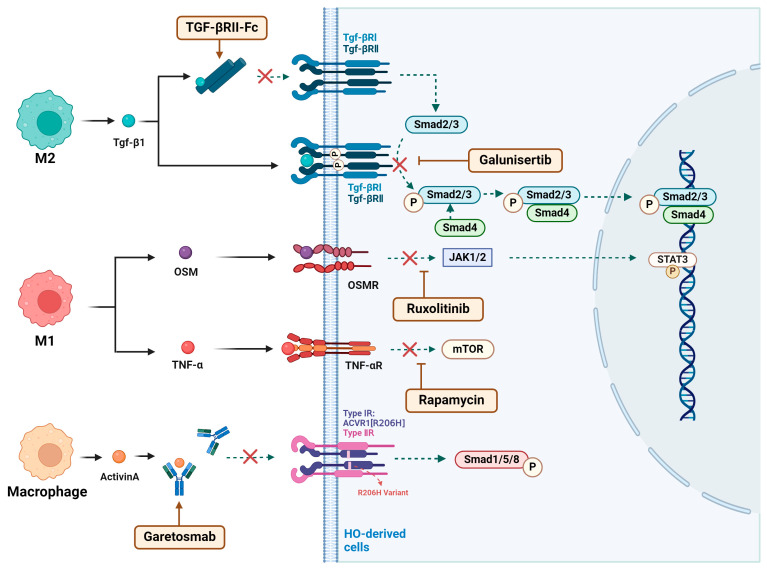
Mechanisms of drugs targeting macrophage-mediated, ossification-related signaling pathways in the treatment of heterotopic ossification. Mechanisms of action of TGF-βRII-Fc, Galunisertib, Ruxolitinib, Garetosmab, and Rapamycin in interfering with macrophage-mediated, ossification-related signaling pathways and their therapeutic potential for HO.

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
