# Peer review of "Macrophage Polarization in Heterotopic Ossification: Inflammation, Osteogenesis, and Emerging Therapeutic Targets"

_ijms, 2025, doi:10.3390/ijms26125821_

Round 1
Reviewer 1 Report
Comments and Suggestions for Authors
The work is well done and organized. I would like to accept the study in present form without any improvement.

Author Response
We sincerely appreciate you taking the time to carefully review our manuscript. We are truly grateful for your support and recognition of our work.
Reviewer 2 Report
Comments and Suggestions for Authors
The author covered the topic of prior research on macrophage role in heterotopic ossification (HO) and included schematic drawings to facilitate the understanding of complex mechanism in this process. They have also stated a lot of possible treatments for HO and described the mechanism of actions. The topic seems relevant, as HO is quite common occurrence in trauma surgery and healing, which can be interesting from a prevention and treatment point of view and as the underlying mechanism is being actively researched. The manuscript is clear, relevant to the field and well structured. The references seem relevant with low number of self-citations. This review is complementing general knowledge regarding heterotopic ossification and the role of the macrophages in it with additional emphasis on immune cells involvement and commitment to the process. Various mechanism have been described, although they are missing the link between macrophages and osteoblasts. Authors should add a paragraph describing the link between monocyte/macrophage lineage and osteoblasts in bone remodeling process with the OPG-RANK-RANKL mechanism. From their review it is not clear how the activation of macrophages and cartilage formation to bone formation and remodeling in HO. The target treatment of HO is described in great detail with numerous treatment options and mechanisms of action.
Line 48 – space before word “Studies”
Line 76 – In the sentence “A recent spatial transcriptomics study…” you have not listed the reference for that study! Do you refer to the Reference No.10 or did you forget to insert an appropriate reference? In case that you refer to the reference 10 please put the reference at the end of this sentence.
Line 115 – space after HO
Line 128 – it seems to be a double space after the reference no.15
Line 129 – maybe put IC in parenthesis, as it is an abbreviation to immune checkpoints, and is mentioned for the first time in the text?!
Line 149 – consider putting all latin phrases in italic (in vivo, in vitro,…) for all other phrases in the text
Line 218 – add space after “FOP” and before (40) reference
Line 221 – add space after “week” and before (24) reference Line 99 – space after “mandible”
Line 313 – Add short description regarding determination and classification of metallopeptidases and metaloproteinases with reference as you introduce 2 different nomenclature (metallopeptidase and metalloporteinase) which are not synonimus.
Line 359 – remove space after (55) reference
Line 373 – remove space after (12) reference
Reviewer 3 Report
Comments and Suggestions for Authors
This manuscript provides a comprehensive overview of the potential role of macrophages in heterotopic ossification (HO). While the review appropriately emphasizes the contribution of M1 (pro-inflammatory) macrophages, it offers only a cursory mention of M2 macrophages without exploring their potential role in HO. This is a significant omission, as M2 macrophages encompass a diverse range of subtypes with potentially distinct and relevant functions in tissue repair and remodeling. A detailed discussion of these M2 subtypes and their possible involvement in HO would greatly enrich the manuscript.
Moreover, the review does not address the potential inhibitory effects of inflammation on the chondrogenic differentiation of stem cells. Given that HO progresses via an endochondral ossification pathway, which necessitates an intermediate cartilage phase, the influence of inflammatory signaling on chondrogenesis is a critical aspect that warrants discussion.
Addressing these points would enhance the depth and completeness of the review, offering a more balanced and mechanistic perspective on the complex roles of macrophage subtypes in HO pathogenesis.
Round 2
Reviewer 3 Report
Comments and Suggestions for Authors
The authors have adequately addressed my concerns.